# Exploring Rehabilitation Provider Experiences of Providing Health Services for People Living with Long COVID in Alberta

**DOI:** 10.3390/ijerph20247176

**Published:** 2023-12-13

**Authors:** Sidney Horlick, Jacqueline A. Krysa, Katelyn Brehon, Kiran Pohar Manhas, Katharina Kovacs Burns, Kristine Russell, Elizabeth Papathanassoglou, Douglas P. Gross, Chester Ho

**Affiliations:** 1Neurosciences, Rehabilitation and Vision, Strategic Clinical Network, Alberta Health Services, Edmonton, AB T5J 3E4, Canadajkrysa@ualberta.ca (J.A.K.); kiran.poharmanhas@albertahealthservices.ca (K.P.M.); russek14@mcmaster.ca (K.R.); papathan@ualberta.ca (E.P.); 2Faculty of Nursing, University of Alberta, Edmonton, AB T6G 1C9, Canada; 3Division of Physical Medicine and Rehabilitation, University of Alberta, Edmonton, AB T6G 2E1, Canada; 4Rehabilitation Research Centre, Faculty of Rehabilitation Medicine, University of Alberta, Edmonton, AB T6G 2G4, Canada; brehon@ualberta.ca (K.B.); dgross@ualberta.ca (D.P.G.); 5Community Health Sciences, Cumming School of Medicine, University of Calgary, Calgary, AB T2N 1N4, Canada; 6School of Public Health, University of Alberta, Edmonton, AB T6G 1C9, Canada; katharina.kovacsburns@albertahealthservices.ca; 7Department of Clinical Quality Metrics, Alberta Health Services, Edmonton, AB T5J 3E4, Canada; 8Department of Physical Therapy, Faculty of Rehabilitation Medicine, University of Alberta, Edmonton, AB T6G 2G4, Canada

**Keywords:** COVID-19, long COVID, provider experience, qualitative

## Abstract

Background: COVID-19 infection can result in persistent symptoms, known as long COVID. Understanding the provider experience of service provision for people with long COVID symptoms is crucial for improving care quality and addressing potential challenges. Currently, there is limited knowledge about the provider experience of long COVID service delivery. Aim: To explore the provider experience of delivering health services to people living with long COVID at select primary, rehabilitation, and specialty care sites. Design and setting: This study employed qualitative description methodology. Semi-structured interviews were conducted with frontline providers at primary care, rehabilitation, and specialty care sites across Alberta. Participants were interviewed between June and September 2022. Method: Interviews were conducted virtually over zoom, audio-recorded, and transcribed with consent. Iterative inductive qualitative content analysis of transcripts was employed. Relationships between emergent themes were examined for causality or reciprocity, then clustered into content areas and further abstracted into a priori categories through their interpretive joint meaning. Participants: A total of 15 participants across Alberta representing diverse health care disciplines were interviewed. Results: Main themes include: the importance of education for long COVID recognition; the role of symptom acknowledgement in patient-centred long COVID service delivery; the need to develop recovery expectations; and opportunities for improvement of navigation and wayfinding to long COVID services. Conclusions: Provider experience of delivering long COVID care can be used to inform patient-centred service delivery for persons with long COVID symptoms.

## 1. Introduction

Coronavirus Disease 2019 (COVID-19), caused by the Severe Acute Respiratory Syndrome Coronavirus 2 (SARS-CoV-2) virus, has had a significant impact on populations worldwide. Samples collected between April to August 2022 for the most recent cycle of the Canadian COVID-19 Antibody and Health Survey demonstrated that over one-half (53.9%) of Canadians had blood antibodies related to a past infection with SARS-CoV-2 [1]. While most who contract COVID-19 will fully recover within three months, a significant proportion experience new or long-lasting symptoms from the virus that persist or present episodically, months to years beyond the initial recovery phase (i.e., long COVID) [2]. Long COVID symptoms can be multisystem and varied, with fatigue, breathlessness, and cognitive dysfunction among the most persistent of symptoms at 1-year follow up [3,4,5]. Individual variation in affected body systems, symptom profiles, and fluctuation in severity of symptoms over time complicates the ability to diagnose, estimate duration, and predict recovery trajectories for long COVID [6,7]. To support recovery, some individuals require little more than self-management resources; while others require more targeted services to treat persistent or relapsing symptoms and resultant loss of function [8,9,10]. Beyond the physical effects, the persistent and variable nature of long COVID symptoms can also impact psychological wellbeing, reducing overall quality of life [11]. Consequently, long COVID management and rehabilitation programs must encompass a wide variety of disciplines to meet the varied needs of patients. Evidence demonstrates that multidisciplinary long COVID rehabilitation and management programs can improve patient functional status and reduce symptom severity by addressing specific patient needs [12,13,14,15]. 

The provider experience is particularly important to understand in the context of care provision for novel conditions, as elaborating these experiences can facilitate identification of challenges in delivering services and opportunities for improvement. Improving provider well-being is a component of the Quadruple Aim, which are organizational aims for quality [16]. Many healthcare delivery organizations use the Quadruple Aim as a guiding framework to optimize health system performance. “Quadruple” refers to the four pillars of the model: enhancing patient experience, improving population health, reducing costs, and improving the work life of health care providers [16]. Provider well-being is linked to patient outcomes, and burnout among healthcare staff is associated with lower quality, less patient-centred care [16]. Currently, there is little known about the provider experience of providing health services for people living with long COVID across diverse care settings [17]. 

The aim of this study was to explore the frontline provider experience of providing health services to people living with long COVID at select primary, rehabilitation, and specialty care sites. Specific objectives were to understand: (1) current approaches to long COVID screening and diagnosis; (2) barriers and enablers of long COVID care provision; and (3) the utility and appropriateness of available long COVID resources and services.

## 2. Method

In this paper, we explore rehabilitation healthcare provider perspectives on Long COVID care delivery. This study received ethics approval from the University of Alberta Health Research Ethics Board (Pro 00113182). 

### 2.1. Study Design

We conducted a qualitative description study involving interviews with healthcare providers who managed people living with long COVID in diverse healthcare settings and geographical regions in Alberta, Canada. This study is part of a broader, mixed-methods study examining the impact of the implementation of a novel long COVID rehabilitation framework that includes exploration of patient perspectives on long COVID services and healthcare utilization of people living with long COVID [18,19]. Qualitative description was selected for its ability to produce a comprehensive, uncomplicated explanation of a phenomena of interest within a healthcare setting [20,21]. 

### 2.2. Setting and Participants

Using purposive sampling, we sought representation from across Alberta, across the care continuum, and across provider disciplines. Provider disciplines included nurses (registered nurses and nurse practitioners) in primary and rehabilitation care; physiotherapists; occupational therapists; pharmacists; and recreational therapists. Inclusion criteria were frontline clinical staff from sites that self-identified as delivering long COVID services; and staff whose day-to-day practice directly involved service delivery for people living with long COVID. Participating sites included primary, rehabilitation, and specialty care sites across Alberta that deliver a wide range of clinical services, including, but not limited to, long COVID services. At the time of the study, there were a limited number of sites providing long COVID management services; invitations to participate were sent to sites known to the research team to be providing these services. 

### 2.3. Recruitment

Clinical managers at participating sites were asked to distribute an invitation email to frontline providers at their site involved in service delivery for people living with long COVID. This email invited providers to contact the research team if they wished to participate in a one-time, virtual interview. One-on-one and group interviews were conducted, per provider preference. These providers received an information letter and consent form prior to the interview. Participants provided written consent, and consent was discussed at the beginning of each interview. The limited number of sites delivering long COVID services and their busy nature constrained recruitment; recruitment of new participants ceased when it was no longer feasible due to capacity of the participating sites.

### 2.4. Question Guide Development

A one-time, virtual co-design session with patient and family advisors with lived experience, clinicians, and administrative, operational, and health system leadership determined thematic areas critical to the experience of navigating long COVID services to inform health service planning; these themes informed the development of the provider interview guide used in the present study. Questions were formulated with input of long COVID care professionals and persons with lived experience of long COVID. A comprehensive description of this co-design process is detailed elsewhere [18]. Questions pertained to: (1) current approaches to screening and assessing for long COVID rehabilitation needs; (2) concomitant challenges and enablers of rehabilitative care provision for people living with long COVID; and (3) utility and appropriateness of available long COVID resources. 

### 2.5. Data Collection

Semi-structured interviews utilizing open-ended questions were conducted with interested providers. Interviews were 30–60 min in length. All interviews were audio-recorded and transcribed verbatim, and written field notes were recorded during the interview. Field notes informed the analysis process prior to and after coding. 

### 2.6. Data Analysis 

All transcripts were analyzed using NVIVO 12 (QSR International Pty Ltd., Burlington, MA, USA) software. Iterative inductive qualitative content analysis was employed to understand the commonalities and differences in provider experience of long COVID service delivery provision and identify opportunities for improvement [21,22,23].

Analysis, which took place over several months, began by one interviewer (SH) and one other member of the research team (KB) independently coding each interview using open coding. A third team member (JAK) independently reviewed the coding to ensure interpretations reflected the data and resolved any discrepancies in coding. Following open coding, codes were compared and contrasted across participants, and then clustered into content areas and further abstracted into a priori categories through their interpretive joint meaning [24]. Relationships between emergent themes were examined for causality or reciprocity, and to produce a holistic understanding of the participants’ experiences. Team members met regularly to review draft themes and further develop the coding framework. An audit trail was kept of decision-making throughout the interview and analysis process. 

## 3. Results

We interviewed 15 frontline healthcare providers from five different healthcare sites in the Calgary, Edmonton, North, and Central zones in Alberta, Canada. Interviews were held between June and September 2022.

Table 1 below presents details on provider interview dates, duration, number of participants, participant professional roles, and group size.

### 3.1. Importance of Patient/Provider Knowledge for Long COVID Recognition

Knowledge of methods of long COVID diagnosis, screening tools, and appropriate patient- and provider-facing resources varied significantly among participants. While most participants were aware of how to screen for long COVID, others were not and relied on other health professionals to identify and diagnose suspected long COVID patients. In these latter cases, lack of education on how to search for and diagnose these patients was cited as the reason for the gap in knowledge.


*When we had all our education on this, that wasn’t really part of the education… searching for long COVID.*
(Primary care provider—Interview 8)

Many participants spoke of the need for adequate awareness and knowledge of long COVID to be able to recognize symptoms and to direct individuals to the appropriate rehabilitation resources and services to manage symptoms.


*(Patients) are not getting the support from anywhere else; a lot of them are really shocked when they talk to their GPs (primary care physician) and the GP isn’t aware that [specialty clinic] is here.*
(Rehabilitation care provider—Interview 1)

Participants described the broad range of known long COVID symptoms as a significant barrier to accurate identification and diagnosis of long COVID.


*When I was reading… (experts) were talking about 200 symptoms of long COVID, I don’t know all 200 symptoms. That just blew me away… oh my gosh, we don’t ask 200 symptoms on our (long COVID) questionnaire.*
(Rehabilitation care provider—Interview 1)

Many participants described how patient awareness of long COVID enhanced individual ability to search for answers regarding long COVID symptoms; providers described how this awareness facilitated identification and treatment of these individuals. 


*Some of the patients that do come to us, they tell us that they read about a case in the newspaper and then they went and told their doctor.*
(Specialty care provider—Interview 6)

As providers learned and became more proficient in managing long COVID, they described increased confidence in supporting those with symptoms and directing them to the appropriate resources and services, when required. 


*My clinical assessments have certainly evolved since being in this since January. Just asking more questions, being more astute… in the long COVID world, it’s things that they’re saying and it’s like let’s move over there and talk about that. So, I think that’s just me evolving in my assessment skills.*
(Specialty care provider—Interview 9)

### 3.2. The Role of Patient/Provider Acknowledgement of Long COVID

#### 3.2.1. Disbelief in COVID or Long COVID 

A prominent theme throughout the interviews was that patient or provider disbelief in COVID and long COVID posed a significant barrier to receiving support for people with long COVID. Participants discussed an observed lack of acceptance and recognition of the importance of long COVID within the community. Consequently, providers found they were often the only source of support for those coming to them seeking help for their symptoms. 


*(Patients) are not getting the support from anywhere else. (I) Just feel that a lot of our (patients), especially the long-haul ones, what we’re calling long COVID, when they’re 12, 14, 16 weeks out. They’re just finding there’s not a lot of buy-in, from friends, from family, from employers… We’re finding there’s not a lot of buy-in from even our medical community, right now.*
(Rehabilitation care provider—Interview 1)

Participants described how some patients struggled to find a physician who believed their symptoms were real. For these patients, it was more difficult to attain appropriate care. Rehabilitation or specialty care providers found this difficult as it impeded their ability to find adequate care for patients in the community, particularly in cases where a referral from a nurse practitioner or physician in primary care was required for the patient to receive more specialized forms of care.


*“It’s not the patient’s fault because they don’t know what’s available to them… And when they do ask for help, I have heard ‘my physician doesn’t believe me, they don’t believe in COVID, they don’t believe I’m experiencing these symptoms, and they don’t know where to send me if I’m experiencing these symptoms’”.*
(Specialty care provider—Interview 6)

#### 3.2.2. Acknowledgement as Part of Care

Several providers identified that acknowledgement of long COVID symptoms is the first and most important step of the patient’s long COVID care journey, and a key component of ensuring that patients feel supported. 


*“I think another advantage (of long COVID care) is, the patients feel supported and that they’re not losing their mind, and it’s all in their head. They’re like… I still feel (bad) and nobody knows, we don’t know ‘cause this is new. So I think that’s a big plus, a big advantage for the patients to feel that, just feel supported”.*
(Primary care provider—Interview 8)

Providers spoke of the relief their patients expressed after finally receiving a diagnosis and support for managing their condition. 


*“I think the support piece has been really big (at our site) and that’s what we hear in feedback is just how grateful they are that somebody is here to listen and believes them”.*
(Rehabilitation care provider—Interview 1)

Participants noted that patients who did not receive acknowledgement of their symptoms from their frontline provider experienced challenges finding resources and supports.


*“Most of my patients have talked to… their primary care provider about their symptoms and they’re told that long COVID is a problem and… they could be experiencing those symptoms, but that… seems to be where the conversation ends. … primarily patients… are saying that… they talked to their provider about their symptoms, but they don’t know what else to do. … And (are) not provided with those additional resources”.*
(Specialty care provider—Interview 7)

### 3.3. Developing Recovery Expectations

Providers generally found the available long COVID services and resources, which include online rehabilitation resources, in-person rehabilitation services, and specialty services, to be beneficial for patients. However, providers described how many patients, especially at the initial visit, were seeking a cure for their symptoms. In these situations, providers stressed the importance of developing realistic treatment goals and recovery expectations with patients.


*“I can only think of one (patient) in particular off the top of my head that did not find the resources helpful, and she was really looking for much more specific (steps)… Some conversation… and some understanding of really what’s appropriate for treatment was helpful for her”.*
(Specialty care provider—Interview 7)

Developing effective expectations involved fostering realistic ideas of what to expect from long COVID rehabilitation, helping patients understand that recovery trajectories are uncertain, that rehabilitation may not cure symptoms, and that effective self-management is a major component of care. 


*“I impart on my patients that I don’t have a pill that I can give you that’s going to get rid of your post COVID symptoms. I have some medications that can help manage some of the symptoms that you may experiencing, but overall, you’ve got to put in the work, I’ll put in the work with you, but you’ve got to put in the work yourself and here are the tools I’m handing to you”.*
(Specialty care provider—Interview 6)

### 3.4. Navigation and Wayfinding

#### 3.4.1. Poor Integration with Social and Mental Health Services

Poor integration of long COVID rehabilitation with social and mental health services was a common challenge identified by providers. Clinicians noted that patients could benefit from a more streamlined approach to providing psychosocial services to people living with long COVID. It was recommended that these services, such as mental health or social work, could be directly integrated into provincial long COVID care teams to improve care. 


*“I would love for us to have… some kind of mental health resource—so either a psychologist or a psychiatrist to refer our patients to because a lot of our patients have mental health needs that were either borderline or nonexistent before COVID, and after their COVID illness have significantly impacted them. And so, when we get one of those, where do you go from here?”*
(Specialty care provider—Interview 6)

#### 3.4.2. Wait Times/Referrals

A referral from a primary care provider (e.g., nurse practitioner or physician) was part of the eligibility criteria for patients to access specialized long COVID care. This posed a challenge for patients without access to a family doctor, and placed additional time burdens on patients and allied health providers as they worked to find a primary care physician from whom the patient could receive the required referral. 


*“Some of those people who don’t connect with a family doctor, fall through the cracks. It delays referrals—that kind of thing, and potentially leads to duplication. So I think that’s an issue, is maybe needing that referral from the family doc for people who don’t have it”.*
(Rehabilitation care provider—Interview 2)

Further navigation challenges were experienced by patients once they received a referral. Wait times for specialist care or certain long COVID rehabilitation clinics was a recognized challenge to meeting patient’s recovery needs in a timely manner. 


*“I would say there’s potential (for patients to access long COVID health services), but I think the wait list is so long and depending on people’s needs, like in a timely manner I would say no… and that is certainly a challenge for us, is it’s just backlogged”.*
(Rehabilitation care provider—Interview 2)

## 4. Discussion

Health care providers involved in our study stressed the importance of patient and provider knowledge to recognize long COVID; the role symptom acknowledgement played in patient-centred long COVID service delivery; and the need to develop recovery expectations and key opportunities of improvement for navigation of long COVID services. 

The emerging themes led to the development of a conceptual representation of provider perceptions of long COVID service delivery (Figure 1). The bidirectional arrows demonstrate the negative and positive provider perceptions of long COVID service delivery; lack of patient or provider knowledge negatively affects all components of the model, and conversely, sufficient knowledge helps move the model positively. Knowledge influences acknowledgement, which is necessary to develop recovery expectations and to inform system navigation. Screening and management, key components of system navigation, are similarly impacted by knowledge, acknowledgment, and recovery expectations. If rehabilitation and primary care providers are not knowledgeable about long COVID, do not acknowledge the condition, and do not manage expectations appropriately, then patients do not successfully receive screening and management for long COVID. Successful navigation and wayfinding advance patient long COVID knowledge, how patients receive this acknowledgement, and development of recovery expectations.

Challenges to provider recognition of long COVID stem from lack of knowledge on how to identify patients and the broad range of symptoms associated with long COVID. As their knowledge on long COVID improved so did their abilities to recognize the symptoms and provide appropriate resources to patients. Symptoms are difficult to definitively attribute to long COVID; in many cases, people living with long COVID report that their symptoms are not taken seriously due to lack of understanding by healthcare provider [25] Ensuring providers are knowledgeable about the broad presentation of long COVID symptoms is imperative to improve patient access to long COVID health services. 

This study highlights the essential role of healthcare providers in long COVID service delivery, specifically in acknowledging and validating patients’ symptoms, and helping to foster realistic recovery expectations The uncertain and ambiguous recovery path seen in Long COVID is a challenge reported in our study and is echoed by patients in other studies [26,27,28,29,30]. Long COVID providers described the feelings of reassurance and relief expressed by their patients when providers acknowledged patient symptoms and the uncertainty of prognosis. These insights offer valuable guidance for healthcare providers to deliver the type of care most appreciated by patients with long COVID. 

Challenges navigating the system were highlighted in our interviews. Participants felt that oftentimes their fellow primary care providers acted as barriers to patients’ access to care. A recent qualitative study (*n* = 24) of patient perceptions of care supports this finding; therein patients emphasized the “hard work” in finding the right care provider, one who believed their symptoms were real [27]. Patient-centered care, a crucial aspect of the quadruple aim of healthcare institutions, emphasizes understanding patient perceptions of care [16]. In the search for answers on persistent COVID symptoms, patient inability to find a knowledgeable, empathetic primary care provider is common [25,31]. In [27] and our study, navigation challenges often resulted in delays in accessing services for long COVID. Provider awareness of long COVID varies widely, which leads to inequity. As long COVID emerges as a novel chronic condition, diagnostic criteria continue to evolve, posing additional challenges to recognition and diagnosis. Clinical criteria from information hubs such as the Public Health Agency of Canada may be unable to keep pace with the rapidly evolving nature of long COVID evidence. Similarly, front-line providers may also be unable to keep abreast of the evolving evidence given competing priorities, diverse information resources, and differing levels of comfort in information-seeking behaviours. Revisiting education and awareness strategies in collaboration with providers, to better reach less-aware providers and to more easily update those in the know may be a key next step to improving access to knowledgeable long COVID service providers. 

Our study highlighted a need to better integrate social and mental health services into long COVID service delivery. A growing body of evidence suggests a high prevalence of social and mental health challenges among people living with long COVID [30]. It is evident that health systems must not only treat the physical and cognitive effects of long COVID, but also the social and mental health impacts of the disorder. Services to treat these conditions must be integrated into long COVID care models to best serve those seeking care. 

### Limitations and Strengths

The study’s strengths included the varied geographic representation of nursing and allied healthcare providers across the care continuum, and the iterative nature of the interview process. Providers represented a diversity of sites both geographically and across the care continuum, including primary care, rehabilitation care, and specialty health services. which allowed for a better understanding of common experiences amongst care providers within these professions across the province. Interviews occurred over several months, which allowed for exploration of emergent themes, deepening the understanding of provider perspectives on long COVID service delivery. 

Our study focused on front-line providers, predominantly nursing and allied health professionals, at self-identifying long COVID care sites. This may limit the representation of physician perspectives. We also did not sample until saturation was reached due to practical reasons. Therefore, the findings and conclusions may not fully capture the full range of healthcare provider perspectives. 

We did not explicitly compare or contrast the experiences of healthcare professionals by discipline, which limits our understanding of how perceptions may vary among different types of care professionals. Future research should consider exploring the impact of professional discipline on long COVID service delivery experiences and perspectives.

## 5. Conclusions and Future Implications

This study fills a key gap in the literature on the perceptions of healthcare providers regarding long COVID rehabilitation service delivery. The findings provide important information on essential components of patient-centred care that can be used to improve current practices of long COVID service delivery. Provider education on long COVID appears crucial for improved recognition of the condition and provision of appropriate, patient-centred service delivery to patients. Challenges in navigating the healthcare system, including delays in accessing care and barriers posed by fellow providers, highlight the need for improved coordination and support, particularly the integration of social and mental health services into current services.

Collaboration amongst healthcare professions is key to ensure that regardless of setting, patients are connected to healthcare providers that can recognize, understand, and provide appropriate care and resources needed for recovery and improved quality of life.

## Figures and Tables

**Figure 1 ijerph-20-07176-f001:**
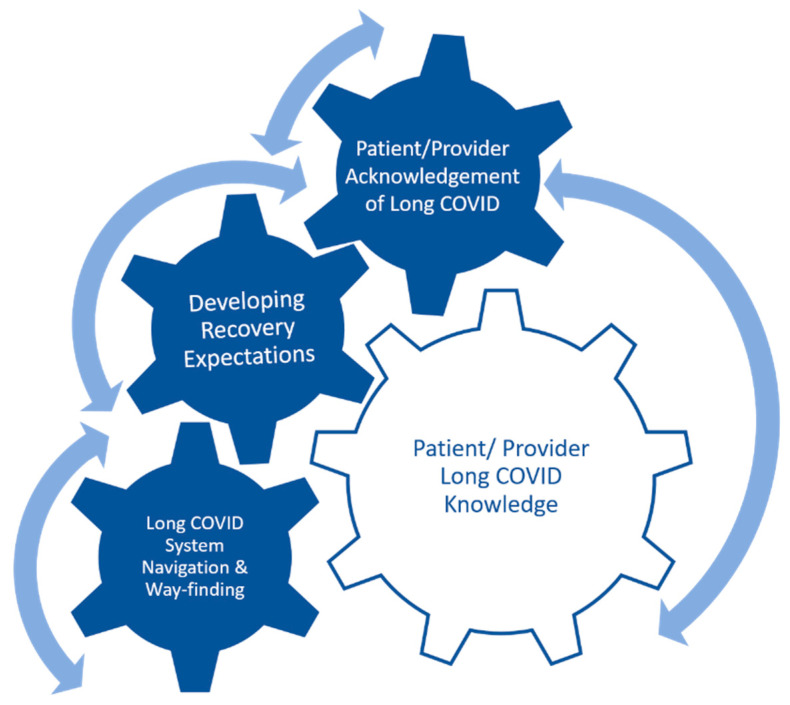
Conceptual framework: Interplay of Long COVID Education, Acknowledgement, Expectations and System Navigation. The complex interplay between the facets of the model is portrayed. The movement of one component of the model sets the others in motion.

**Table 1 ijerph-20-07176-t001:** Frontline Provider Interview/Focus Group Demographics.

Variable	Participants (*n* = 15)
Interview Duration (range, minutes)	20–60 min
Participants (*n*)	15
1:1 Interviews (*n*)	5
Group Interviews (*n*)	2
Group Interviews (range)	2–8 participants
Types of Participant Professional Roles	Registered Nurse, Physiotherapist, Nurse Practitioner, Occupational Therapist, Pharmacist, Recreation Therapist

## Data Availability

The data presented in this study are available on request from the corresponding author. The data are not publicly available due to participant confidentiality.

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
