# Peer review of "Exploring Rehabilitation Provider Experiences of Providing Health Services for People Living with Long COVID in Alberta"

_ijerph, 2023, doi:10.3390/ijerph20247176_

Round 1
Reviewer 1 Report
Comments and Suggestions for Authors
Thanks for giving me chance to review this article.
The sample size is very small(n=15), is the statistical power being conducted in this research?
thanks!
Comments on the Quality of English Languagenil
Author Response
As this is a qualitative study, no statistical power was calculated. n=15 is within the normal range of participants in qualitative research.
Reviewer 2 Report
Comments and Suggestions for Authors
The aim of this study was to explore the frontline provider experience of providing health services to people living with long COVID at select primary, rehabilitation, and speciality care sites. Specific objectives were to understand: 1) current approaches to long COVID screening and diagnosis; 2) barriers and enablers of long COVID care provision; and 3) the utility and appropriateness of available long COVID resources and services.
Abstract: Please shorten the background. Make a section called Participants where you include the participant information.
Introduction: L47-48: Please remove underline.
L80: Place "and" before 3)
Methods: Please provide participant information.
Question Guide sample must be provided in the appendix.
L133: NVIVO 12 software (Company, Country)
Table 1: Please provide participant information (age, gender etc.)
Conclusion and Future Implications: Please make it substantially shorter.
Comments on the Quality of English Language
Some minor proof reading needed.
Author Response
Abstract: Please shorten the background. Make a section called Participants where you include the participant information.
-Removed sentences from the background and added additional participant information
Introduction: L47-48: Please remove underline.
-Removed underline.
L80: Place "and" before 3)
- placed and before 3.
Methods: Please provide participant information.
-added more information regarding the participants locations.
Question Guide sample must be provided in the appendix.
- My colleague will provide a question guide sample to the editor.
L133: NVIVO 12 software (Company, Country)
- added this information: QSR International Pty Ltd., USA
Table 1: Please provide participant information (age, gender etc.)
- we did not collect age or gender information from participants.
Conclusion and Future Implications: Please make it substantially shorter.
- significantly shortened these sections to make more concise.
Reviewer 3 Report
Comments and Suggestions for Authors
Thank you for the opportunity to review this manuscript. In the abstract the authors refer to the themes as 'Importance of education for long COVID recognition; the role of symptom acknowledgement in patient-centred long COVID service delivery; the need to develop recovery expectations; and opportunities for improvement of navigation and wayfinding to long COVID Services.' which are different to those within the main text. Perhaps the authors could align them more closely.
I am suggesting that the authors align the titles for the themes which are described in the abstract with the titles for the themes within the body of the manuscript as they differ that is all.
Author Response
Thank for your comment, there was an issue with the sub-theme formatting that I have corrected. There are now 4 high-level themes that reflect those identified in the abstract. I have also slightly edited the titles to more accurately reflect the abstract.
Reviewer 4 Report
Comments and Suggestions for Authors
Line 47-49: Why underline?
Line 96: Though purposive sampling was used, a description of how the sites were selected will show how bias in site selection was addressed.
Recruitment: Provide information on response rate, if available.
Line 113: Typo (sites)
The methodology was thorough and well-detailed. Steps taken to minimize bias in the coding of responses were described.
Lines 149 and 363: Providing more information about the geographic spread from where participants were recruited could justify the claim of geographic diversity, which was stated as a strengths section.
Line 153: What is the number of each healthcare provider represented?
Overall, a well-written study with a good background and justification. Study questions were answered by the results section and the analytical method was appropriate for the study design.
Author Response
Line 47-49: Why underline?
- typo; removed underline.
Line 96: Though purposive sampling was used, a description of how the sites were selected will show how bias in site selection was addressed.
- added sentence regarding site selection
Recruitment: Provide information on response rate, if available.
- not available; managers sent invitations to team members, but did not disclose the number of invitations. Response rates are not normally recorded in qualitative interview studies.
Line 113: Typo (sites)
- fixed
What is the number of each healthcare provider represented?
- because there are few sites providing this type of care in the province, we have avoided providing the numbers of health care providers due to the potential for identification, as some professions had an n of 1.
Reviewer 5 Report
Comments and Suggestions for Authors
It would be beneficial to describe the interview's development and foundation. Does the relevant questionnaire utilize a confirmed method? If so, how are the questions chosen?
Author Response
It would be beneficial to describe the interview's development and foundation. Does the relevant questionnaire utilize a confirmed method? If so, how are the questions chosen?
added a line to describe the background of the question guide.
I have also moved the information regarding the questions to the other information about the question guide to more clearly describe the background.
The confirmed method used is the qualitative interview.
Round 2
Reviewer 1 Report
Comments and Suggestions for Authors
nil